# Ubiquitin-interacting motifs of ataxin-3 regulate its polyglutamine toxicity through Hsc70-4-dependent aggregation

Sean L Johnson[1], Bedri Ranxhi[1], Kozeta Libohova[1], Wei-Ling Tsou[1]*, Sokol V Todi[1,2]*

[1]Department of Pharmacology, Wayne State University, Detroit, United States; [2]Department of Neurology, Wayne State University, Detroit, United States

**Abstract** Spinocerebellar ataxia type 3 (SCA3) belongs to the family of polyglutamine neurodegenerations. Each disorder stems from the abnormal lengthening of a glutamine repeat in a different protein. Although caused by a similar mutation, polyglutamine disorders are distinct, implicating non-polyglutamine regions of disease proteins as regulators of pathogenesis. SCA3 is caused by polyglutamine expansion in ataxin-3. To determine the role of ataxin-3's non-polyglutamine domains in disease, we utilized a new, allelic series of *Drosophila melanogaster*. We found that ataxin-3 pathogenicity is saliently controlled by polyglutamine-adjacent ubiquitin-interacting motifs (UIMs) that enhance aggregation and toxicity. UIMs function by interacting with the heat shock protein, Hsc70-4, whose reduction diminishes ataxin-3 toxicity in a UIM-dependent manner. Hsc70-4 also enhances pathogenicity of other polyglutamine proteins. Our studies provide a unique insight into the impact of ataxin-3 domains in SCA3, identify Hsc70-4 as a SCA3 enhancer, and indicate pleiotropic effects from HSP70 chaperones, which are generally thought to suppress polyglutamine degeneration.

*For correspondence:
wtsou@wayne.edu (W-LT);
stodi@wayne.edu (SVT)

**Competing interests:** The authors declare that no competing interests exist.

## Introduction

Spinocerebellar ataxia type 3 (SCA3; also known as Machado-Joseph disease) is the most frequent dominant ataxia worldwide. SCA3 is caused by CAG repeat expansion in the gene *ATXN3* that is normally 12–42 repeats long but is expanded to ~60–87 repeats in patients (*Todi et al., 2007b*; *Costa and Paulson, 2012*). This triplet repeat encodes a polyglutamine (polyQ) tract in the protein, ataxin-3, a deubiquitinase (DUB; *Figure 1*) implicated in protein quality control and DNA repair (*Costa and Paulson, 2012*; *Dantuma and Herzog, 2020*). The precise molecular details of SCA3 pathogenesis remain unclear.

SCA3 is one of the nine neurodegenerative disorders caused by abnormal polyQ expansion in diverse proteins (*Todi et al., 2007b*; *Costa and Paulson, 2012*). The polyQ family includes Huntington's disease (HD), SCAs 1, 2, 3, 6, 7, and 17, Dentatorubral–pallidoluysian atrophy, and Kennedy's disease. Together, HD and SCA3 are the two most common among the nine known polyQ diseases. Although each polyQ disorder is caused by the aberrant lengthening of the same domain, they are clinically distinct diseases (*Todi et al., 2007b*; *Costa and Paulson, 2012*; *Nath and Lieberman, 2017*; *Pérez Ortiz and Orr, 2018*; *Lieberman et al., 2019*; *Klockgether et al., 2019*). Fundamental to polyQ disorders is the role of protein context, i.e. regions outside of the polyQ that modulate the properties of the elongated tract (*Todi et al., 2007b*; *Nath and Lieberman, 2017*; *Pérez Ortiz and Orr, 2018*; *Lieberman et al., 2019*; *Klockgether et al., 2019*; *Blount et al., 2014*; *Ristic et al., 2018*; *Sutton et al., 2017*; *Tsou et al., 2015a*). It is the protein context that differentiates polyQ diseases; why, for example, expansions in ataxin-3 cause SCA3 instead of another disorder. While studies of other polyQ diseases provided insight into the role of protein context in polyQ

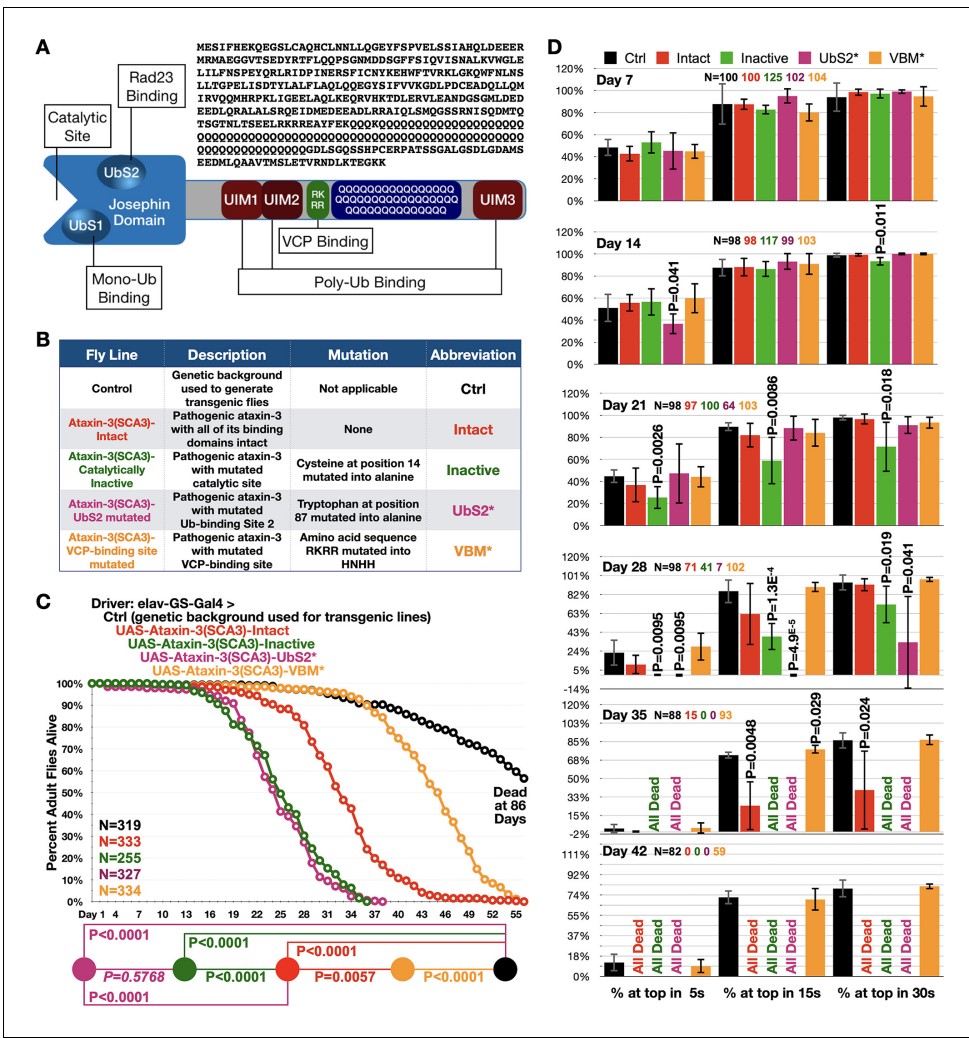

**Figure 1.** Ataxin-3 domains impact its polyQ toxicity. (**A**) Diagram of ataxin-3. Josephin: deubiquitinase domain necessary for ataxin-3's ability to cleave ubiquitin bonds (*Winborn et al., 2008*). UbS: ubiquitin-binding site (*Nicastro et al., 2010*; *Nicastro et al., 2009*); UIM: ubiquitin-interacting motif (*Winborn et al., 2008*); RKRR: amino acid sequence necessary for ataxin-3 to bind directly to VCP, known as VCP-binding motif, VBM (*Boeddrich et al., 2006*). QQQ: polyQ that is expanded in SCA3 (*Costa and Paulson, 2012*). Also shown is the amino acid sequence of human ataxin-3 used. (**B**) Summary of transgenic fly lines, mutations, and abbreviations. (**C**) Longevity results when pathogenic ataxin-3 versions are expressed pan-neuronally in adults. P values: log-rank tests with Holm-Bonferroni adjustment. Italicized P value: not statistically significant. (**D**) Means -/+ SD from negative geotaxis tests. Groups are the same as in (**C**). P values: one-tailed, Student's t-tests comparing lines to control flies of the same day and time point. 'One-tailed' tests were chosen based on the outcomes from longevity data in panel (**C**), which indicated specific directionality in expectations. If no P value is noted, it was equal to or greater than 0.05. Exact P values are given in other cases. The total number of flies in each group is shown in panels. For panel (**D**), the number of independent groups of flies per genotype was: Ctrl: 5, Intact: 5, Inactive: 6, UbS2*: 5, VBM*: 5. Repeats were biological replicates.

The online version of this article includes the following source data and figure supplement(s) for figure 1:

**Source data 1.** Data related to *Figure 1* - Longevity and motility results.

**Figure supplement 1.** Ataxin-3 domains impact the toxicity of its polyQ-expanded variant in glia.

neurodegeneration (*Todi et al., 2007b*; *Nath and Lieberman, 2017*; *Pérez Ortiz and Orr, 2018*; *Lieberman et al., 2019*; *Klockgether et al., 2019*), the protein context for SCA3 has not been clarified. For instance, in SCA1, the nuclear localization signal directs its disease protein, ataxin-1, to the nucleus and is necessary for pathogenesis (*Pérez Ortiz and Orr, 2018*; *Klement et al., 1998*). Studies on SCA7 also identified interactions with other proteins that show a role for non-polyQ domains

in the toxicity of its disease protein, ataxin-7 (*Nath and Lieberman, 2017*; *Lieberman et al., 2019*; *Duncan et al., 2013*). Another good case of polyQ protein biology and expression pattern being connected to phenotype is Kennedy's disease. In this disorder, polyQ expansion occurs near the amino terminus of the androgen receptor (AR), which results in a partial loss of AR function that leads to androgen insensitivity, as well as proteotoxicity. AR is expressed in motor neurons of the spinal cord and the brainstem, where it mediates neurotrophic responses. It is these neurons that degenerate in Kennedy's disease (*Todi et al., 2007b*; *Lieberman et al., 2019*; *Giorgetti et al., 2016*; *Lieberman et al., 2002*; *Lieberman et al., 2014*).

To understand the role of protein context in SCA3, we focused on the domains of ataxin-3. Ataxin-3 contains a ubiquitin-protease (Josephin) domain on its N-terminal half (*Costa and Paulson, 2012*; *Dantuma and Herzog, 2020*; *Figure 1A*). Within it are the catalytic triad that enables ataxin-3 to cleave ubiquitin bonds, as well as two ubiquitin-binding sites (UbS) that either interact with ubiquitin (UbS1), or with ubiquitin and the proteasome-associated protein, Rad23 (UbS2; *Costa and Paulson, 2012*; *Dantuma and Herzog, 2020*). Downstream are two ubiquitin-interacting motifs (UIMs 1 and 2), followed by a site that binds the AAA ATPase, VCP (VCP-binding motif; VBM), and the polyQ tract (*Costa and Paulson, 2012*; *Dantuma and Herzog, 2020*). At the end resides a third UIM (UIM3) (*Costa and Paulson, 2012*; *Dantuma and Herzog, 2020*). In another, less common iso-form UIM3 is replaced by a hydrophobic C-terminus (*Johnson et al., 2019*; *Harris et al., 2010*; *Kawaguchi et al., 1994*). UIMs enable ataxin-3 to bind poly-ubiquitin (*Winborn et al., 2008*; *Todi et al., 2009*; *Todi et al., 2010*).

We previously examined the role of three domains of ataxin-3 in its pathogenicity: UbS2, VBM, and the catalytic site. UbS2 regulates ataxin-3 turnover and toxicity (*Blount et al., 2014*; *Sutton et al., 2017*; *Tsou et al., 2015a*); the VBM controls ataxin-3 aggregation and toxicity, but not turnover (*Blount et al., 2014*; *Ristic et al., 2018*; *Tsou et al., 2015a*); and the catalytic site is necessary for ataxin-3's inherent neuroprotective functions (*Sutton et al., 2017*; *Tsou et al., 2015a*; *Tsou et al., 2013*). These findings led us to examine the role of other domains of ataxin-3 by using novel, isogenic lines of the model organism, *Drosophila melanogaster*. Here, we show that toxicity from ataxin-3's polyQ tract is markedly impacted by the UIMs. Whereas the isolated polyQ is decid-edly pathogenic, UIM addition enhances toxicity. Mutating the UIMs of full-length, pathogenic ataxin-3 renders it less harmful. UIMs interact with heat shock protein cognate 4 (Hsc70-4), which enhances ataxin-3 aggregation and toxicity in a manner dependent on these ubiquitin-binding domains. Additional studies indicate that Hsc70-4 enhances the toxicity of other polyQ proteins and provide a comprehensive view of the relative impact of various ataxin-3 domains on its pathogenic-ity. We introduce unique genetic tools to understand SCA3 biology and to discover and optimize therapeutic options for it. Our studies provide novel insight into protein context in SCA3 and high-light Hsc70-4 as a new target for intervention.

## Results

### The modulatory role of ataxin-3's domains on its toxicity

Protein context is a key determinant of polyQ degeneration (*Todi et al., 2007b*; *Costa and Paulson, 2012*; *Dantuma and Herzog, 2020*; *Nath and Lieberman, 2017*; *Pérez Ortiz and Orr, 2018*; *Lieberman et al., 2019*; *Klockgether et al., 2019*; *Giorgetti et al., 2016*; *Lieberman et al., 2014*), but its importance in SCA3 has not been investigated systematically. We previously examined the role of non-polyQ domains of ataxin-3, UbS2 (*Blount et al., 2014*; *Sutton et al., 2017*; *Tsou et al., 2015a*), and VBM (*Ristic et al., 2018*; *Figure 1A*). Prior work from us, spearheaded by pioneering studies from the Bonini lab, also showed that the catalytic site of ataxin-3 is important for toxicity (*Sutton et al., 2017*; *Tsou et al., 2015a*; *Tsou et al., 2013*; *Warrick et al., 2005*). However, the con-tribution of different domains to ataxin-3 pathogenicity has not been investigated side-by-side.

Using isogenic fly lines (*Figure 1B*), we examined the toxicity of pathogenic ataxin-3 when expressed pan-neuronally in adults using the binary, Gal4-UAS system of expression (*Brand and Per-rimon, 1993*; *Brand et al., 1994*) and a driver that requires the drug, RU486 to induce transgenes (*Nicholson et al., 2008*; *Osterwalder et al., 2001*; *Roman and Davis, 2002*; *Sujkowski et al., 2015*). We selected this approach as SCA3 is adult-onset and progressive and also because neurons are the type of cell impacted. We compared the following full-length, human ataxin-3 versions: intact

domains, catalytically inactive, mutated UbS2 (UbS2* *Blount et al., 2014*; *Sutton et al., 2017*; *Tsou et al., 2015a*), and mutated VBM (VBM* *Ristic et al., 2018*; *Figure 1B*). The polyQ repeat is 78–80, within-patient range (*Costa and Paulson, 2012*). Each transgene is inserted into the same chromosomal location, attP2, as one copy and in the same orientation, leading to similar expression (*Ristic et al., 2018*; *Tsou et al., 2015b*; *Tsou et al., 2016*; *Blount et al., 2018*; *Sutton et al., 2017*; *Johnson et al., 2019*; *Groth et al., 2004*).

Adult, pan-neuronal expression of pathogenic ataxin-3 with intact domains caused early lethality (*Figure 1C*). Catalytically inactive ataxin-3 led to significantly shorter lifespan, as did the version with mutated UbS2. Mutating the VBM led to markedly longer lifespan compared to 'intact'. Still, flies expressing pathogenic ataxin-3 with mutated VBM did not reach normal longevity (*Figure 1C*). These data were complemented by negative geotaxis assays (*Figure 1D*). We observed reduced motility with flies expressing pathogenic ataxin-3 'intact' and further impaired motility with those expressing inactive ataxin-3 or ataxin-3 with mutated UbS2. On the other hand, pan-neuronal expression of pathogenic ataxin-3 with mutated VBM did not significantly impact motility.

Next, we examined the effect of the same ataxin-3 transgenes in glial cells to obtain information on toxicity from pathogenic ataxin-3 in this cell type. Overall trends were similar to neuronal cells: catalytically inactive and UbS2-mutated ataxin-3 led to increased toxicity compared with pathogenic ataxin-3 with intact domains. But, unlike in neurons, in glial cells we did observe a statistical difference in toxicity between pathogenic ataxin-3 with mutated catalytic site and the one with mutated UbS2. There was no statistical difference between controls not expressing pathogenic ataxin-3 and flies expressing pathogenic ataxin-3 with mutated VBM both in terms of longevity (*Figure 1—figure supplement 1A*) and motility (*Figure 1—figure supplement 1B*). These findings highlight a need to explore the role of glia in SCA3.

Based on these collective data, we conclude that domains outside of ataxin-3's polyQ play significant modulatory roles in pathogenicity. These outcomes led us to ask: what is the role of the most common domain of ataxin-3, the UIM, in SCA3?

## UIMs enhance toxicity of the isolated polyQ of ataxin-3

To conduct an initial examination of ataxin-3 UIMs in SCA3, we synthesized transgenes that encode its isolated polyQ by itself or with the sequential addition of UIMs, including a full-length version (*Figure 2A*). We generated isogenic lines that integrated each transgene into locus attP2, same as flies in *Figure 1*. The polyQ for these new flies is encoded by alternating CAGCAA repeats. Long, pure CAG tracts can lead to mRNA-based toxicity (*Li et al., 2008*) and unconventional translation of non-polyQ frames (*Bañez-Coronel et al., 2015*; *Zu et al., 2011*; *Zu et al., 2018*). Our approach mitigates both possibilities (*Li et al., 2008*; *Figura et al., 2015*; *Green et al., 2016*; *Martí, 2016*; *Shieh and Bonini, 2011*; *Sobczak et al., 2003*; *Sobczak and Krzyzosiak, 2005*; *Stochmanski et al., 2012*), allowing us to focus solely on toxicity from the polyQ and specific domains. To the best of our knowledge, ours is the first instance that this approach is undertaken to dissect in detail the role of regions surrounding a polyQ in an intact organism.

The isolated polyQ80 was more toxic than full-length, pathogenic ataxin-3 when expressed in all tissues or only in neurons (*Figure 2B,C*). This pattern was reversed in glia; full-length ataxin-3 was more problematic than polyQ80 in these cells, highlighting a need to better understand glial response to polyQ proteins. Intriguingly, polyQ80 was generally less toxic than the same peptide with appended UIMs when expressed everywhere, selectively in neurons, or in glia; addition of UIM3 appeared particularly pathogenic (*Figure 2B,C*). Eye-restricted expression of these transgenes also led to variable toxicity, with UIM3 addition being more problematic than the isolated polyQ80 or polyQ80 with all UIMs (*Figure 2D*).

Gal4 drivers above enable transgene expression during development and in adults. Because, as mentioned above, SCA3 is adult-onset, we also investigated the effect of polyQ80 without and with UIMs when expressed pan-neuronally only in adults. PolyQ80 was noticeably less toxic than versions with UIMs (*Figure 2E*). Flies expressing Q80-UIM3 or UIM1-2-Q80 lived longer than those expressing polyQ80 with all UIMs (*Figure 2E*).

There were tissue-dependent variations in toxicity from the above constructs. For example, Q80-UIM3 was markedly more toxic in fly eyes compared to the polyQ80 alone or with the addition of other UIMs (*Figure 2D*), whereas polyQ80 with all three UIMs was the most toxic species in adult neurons (*Figure 2E*). These are intriguing findings that require future attention; also, they are not

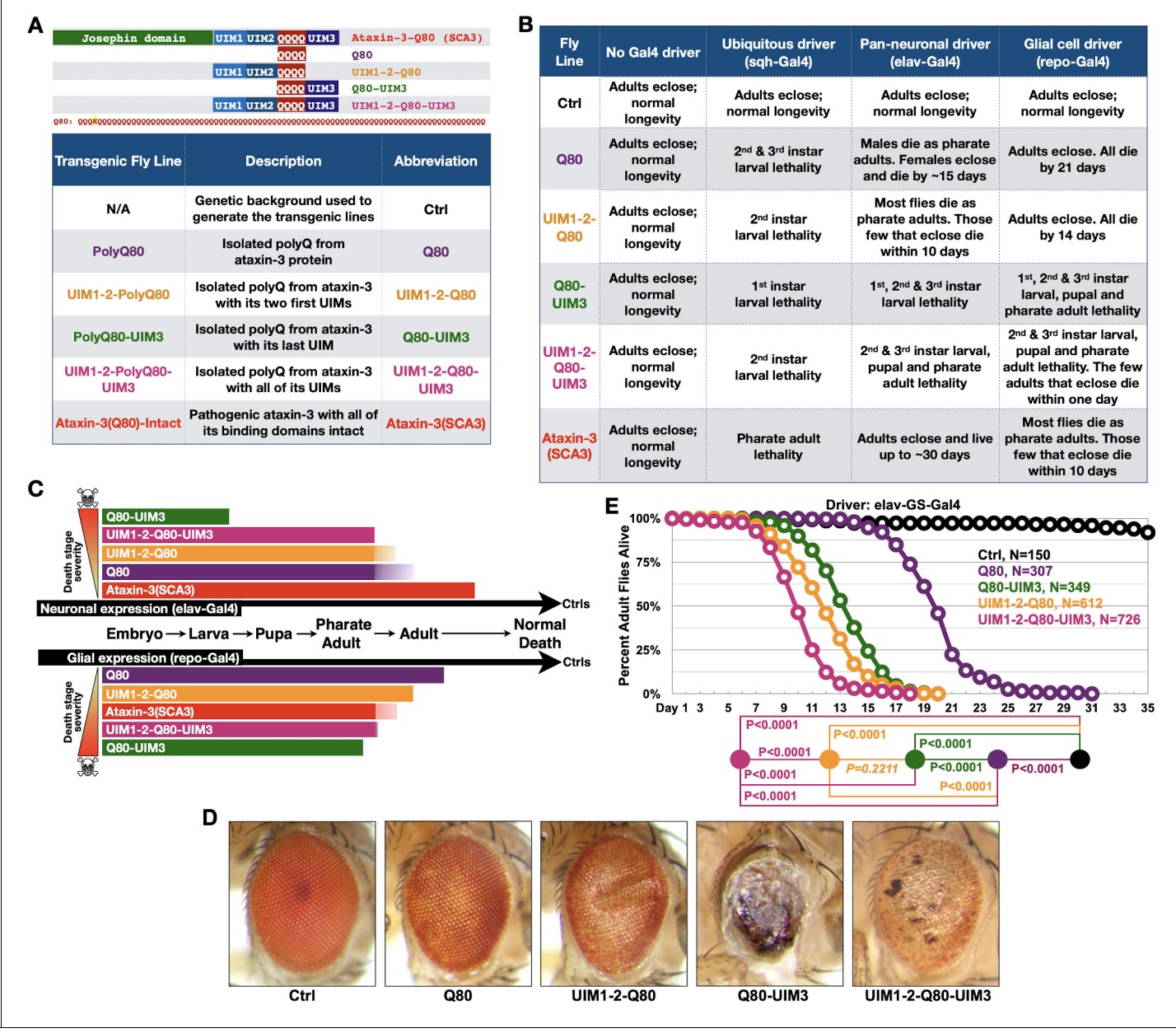

**Figure 2.** UIMs modulate toxicity of the isolated polyQ of ataxin-3. (**A**) Diagram (top) and abbreviations (bottom) for transgenes used. The polyQ of ataxin-3 has a lysine interruption after the first three glutamines (QQQKQQQQ...), highlighted in yellow. (**B**) Summary of results when the noted transgenes are expressed in fly tissues. 1st, 2nd, and 3rd instars are progressive larval stages as flies develop from embryos to pupae. Results in table are representative of independent crosses conducted at least five times each. (**C**) Graphical summary of results from (**B**). Solid color denotes primary observations. (**D**) Photos of adult eyes expressing transgenes through GMR-Gal4. Photos are representative of independent crosses conducted at least five times. Protein levels are shown in *Figure 2—figure supplement 1*. (**E**) Longevity of flies expressing transgenes pan-neuronally in adults. P values: log-rank tests with Holm-Bonferroni adjustment. Italicized P value: not statistically significant.

The online version of this article includes the following source data and figure supplement(s) for figure 2:

**Source data 1.** Data related to *Figure 2* - Longevity results.
**Source data 2.** Data related to *Figure 2* - Uncropped images.
**Figure supplement 1.** Levels of polyQ proteins, related to *Figure 2D*.

dissimilar from the neuronal-glial differences we observed in *Figure 1*, *Figure 1—figure supplement 1*, and *Figure 2B*. These outcomes highlight the utility of the new fly models that we have generated to understand tissue-selective toxicity in vivo in the future. Collectively, data in *Figure 2* indicate a regulatory role for UIMs in the pathogenicity of the expanded polyQ of ataxin-3, and led us to explore them in the full-length protein.

## UIMs of full-length ataxin-3 regulate toxicity and aggregation

To investigate a regulatory role for the UIMs in the pathogenicity of full-length ataxin-3, we generated an additional transgenic line that expresses human ataxin-3 with mutated UIMs (*Figure 3A*). This line was created through site-directed mutagenesis of the plasmid used to generate flies in

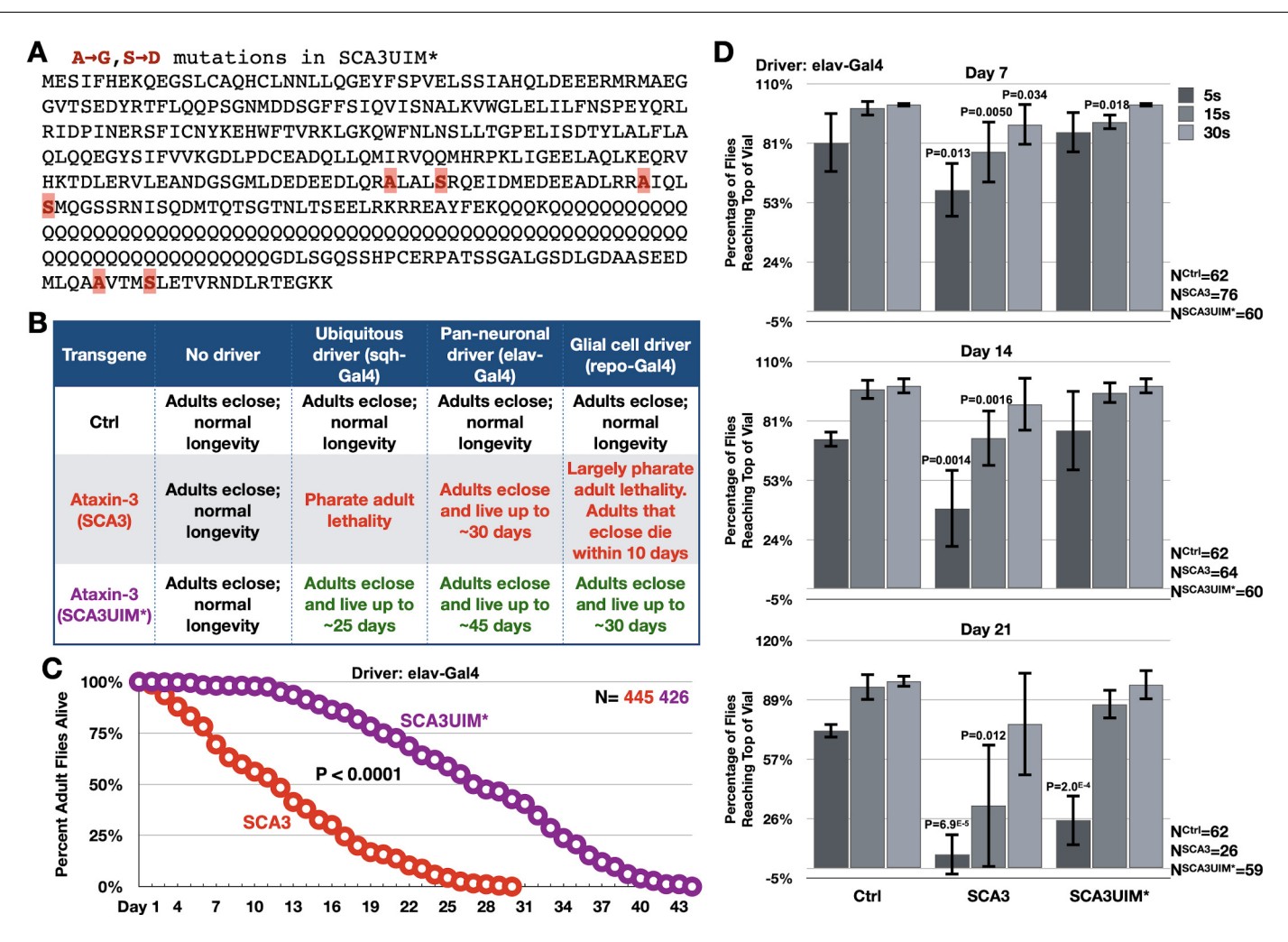

**Figure 3.** Mutating the UIMs of full-length, pathogenic ataxin-3 decreases its toxicity. (A) Amino acid sequence of full-length, pathogenic ataxin-3 and UIM mutations. (B) Summary of results when ataxin-3 flies, or non-expressing controls, are monitored during development and in adulthood. Results are representative of independent crosses conducted at least five times each. (C) Longevity of flies expressing transgenes pan-neuronally, throughout development and in adults. P value: log-rank test. (D) Negative geotaxis assays of adults expressing pathogenic ataxin-3 and their non-expressing controls. Shown are means -/+ SD. P values: two-tailed, Student's t-tests comparing flies at each time point to their corresponding, non-expressing controls. We selected 'two-tailed' tests since we did not know how 'UIM*' would perform compared to 'Ctrl'. If no P value is noted, it was equal to or greater than 0.05. Exact P values are given in other cases. The number of groups of flies tested per genotype was: Ctrl: 6, SCA3: 8, SCA3UIM*: 6. Repeats were biological replicates.

The online version of this article includes the following source data for figure 3:

**Source data 1.** Data related to *Figure 3* - Longevity and motility results.

*Figure 2A*. The mutations that we selected were previously shown to disable the ability of ataxin-3 UIMs to bind poly-ubiquitin (*Winborn et al., 2008*; *Todi et al., 2009*; *Todi et al., 2010*).

Expression of pathogenic ataxin-3 with mutated UIMs was consistently and markedly less toxic than its counterpart with intact UIMs in all of the tissues tested (*Figure 3B,C*). Motility assays also showed that adults expressing UIM-mutated ataxin-3 retained mobility longer than adults expressing pathogenic ataxin-3 with intact UIMs (*Figure 3D*). We conclude that mutating its UIMs renders ataxin-3 significantly less pathogenic.

Next, we examined if reduced toxicity from UIM mutations reflected changes in ataxin-3 levels or aggregation. Mutating the UIMs does not cause loss of ataxin-3 protein (*Figure 4A*). We used three antibodies for this assay: a rabbit monoclonal antibody that recognizes the C-terminal HA tag; a mouse monoclonal antibody that recognizes an epitope upstream of UIM1-2 (1H9); and a rabbit polyclonal antibody generated against full-length ataxin-3 (Machado-Joseph disease [MJD]; *Paulson et al., 1997*). While there is variance in the exact species recognized by each antibody, overall we conclude that mutating the UIMs does not deplete ataxin-3 protein. Differences in species likely stem from epitopes recognized by each antibody, SDS solubility of different ataxin-3 species, and epitope exposure on PVDF membrane.

Prior work showed that nuclear presence of polyQ disease-causing proteins is a critical regulator of pathogenicity (*Lieberman et al., 2019*; *Klement et al., 1998*; *Bichelmeier et al., 2007*; *Macedo-Ribeiro et al., 2009*). Based on subcellular fractionation, there was no significant difference in distribution between ataxin-3 with intact or mutated UIMs (*Figure 4B*). However, according to two different assays, mutating the UIMs rendered pathogenic ataxin-3 less aggregation-prone. Centrifugation-based protocols and filter-trap assays both showed higher aggregation of pathogenic ataxin-3 with intact UIMs (*Figure 4C,D*). PolyQ protein aggregation is a hallmark of this family of diseases (*Todi et al., 2007b*; *Costa and Paulson, 2012*; *Nath and Lieberman, 2017*; *Lieberman et al., 2019*; *Klockgether et al., 2019*; *Paulson et al., 1997*). In our *Drosophila* studies, aggregation of pathogenic ataxin-3 and other polyQ proteins precedes toxicity; also, the level of aggregation mirrors the extent of their pathogenicity (*Ristic et al., 2018*; *Sutton et al., 2017*; *Tsou et al., 2015a*; *Johnson et al., 2019*; *Tsou et al., 2013*; *Tsou et al., 2015b*; *Tsou et al., 2016*). Based on data from *Figures 3* and *4*, we conclude that the UIMs of pathogenic ataxin-3 enhance its aggregation and toxicity in *Drosophila*.

## UIMs mediate ataxin-3 interaction with Hsc70-4

UIMs exist in proteins with various functions and interact with poly-ubiquitin (*Dantuma and Herzog, 2020*; *Andersen et al., 2005*; *Marmor and Yarden, 2004*; *Zheng and Shabek, 2017*; *Dikic et al., 2009*). The UIMs of ataxin-3 indeed bind poly-ubiquitin (*Winborn et al., 2008*; *Todi et al., 2009*; *Todi et al., 2010*). Still, we wondered whether proteins other than ubiquitin also interact with ataxin-3 through UIMs. After all, at least one other ubiquitin-binding domain on ataxin-3, UbS2, interacts with ubiquitin and a non-ubiquitin protein, Rad23 (*Blount et al., 2014*; *Sutton et al., 2017*; *Nicastro et al., 2010*; *Nicastro et al., 2009*; *Nicastro et al., 2005*) To find proteins that interact with ataxin-3 through its UIMs, we homogenized flies expressing ataxin-3 pan-neuronally and depleted resulting lysates of poly-ubiquitin by using tandem ubiquitin binding entities (TUBEs; Methods). Afterward, we immunopurified (IP-ed) ataxin-3 from the supernatants and resolved all proteins on an SDS-PAGE gel (*Figure 5A*). We observed a band above ataxin-3 that was present with intact UIMs but not as clearly visible when they were mutated. Mass spectrometry from the dissected regions yielded heat shock protein cognate 4 (Hsc70-4) only in the presence of ataxin-3 with intact UIMs (*Figure 5A*).

We confirmed mass spectrometry data with co-IPs. Full-length, pathogenic ataxin-3 co-IP-ed endogenous Hsc70-4 in a UIM-dependent manner: whereas pathogenic ataxin-3 with intact UIMs co-IP-ed Hsc70-4, we did not observe the same with UIM-mutated, pathogenic ataxin-3 (*Figure 5B*). The interaction of ataxin-3 with Hsc70-4 could potentially rely more heavily on UIM3. We inferred this from IPs with isolated polyQ peptides. PolyQ80-UIM3 co-IP-ed endogenous Hsc70-4 more readily than polyQ80 alone or polyQ80 with UIMs 1 and 2 (*Figure 5C*). In panel 5D, we conducted co-IPs with modified conditions to approximate more the levels of the HA-tagged polyQ forms. All three polyQ80 versions co-IP-ed Hsc70-4, but UIM3 appeared more important than the two other UIMs in facilitating an interaction. Collectively, these results highlight the UIMs of ataxin-3 as an interaction site for Hsc70-4 in vivo.

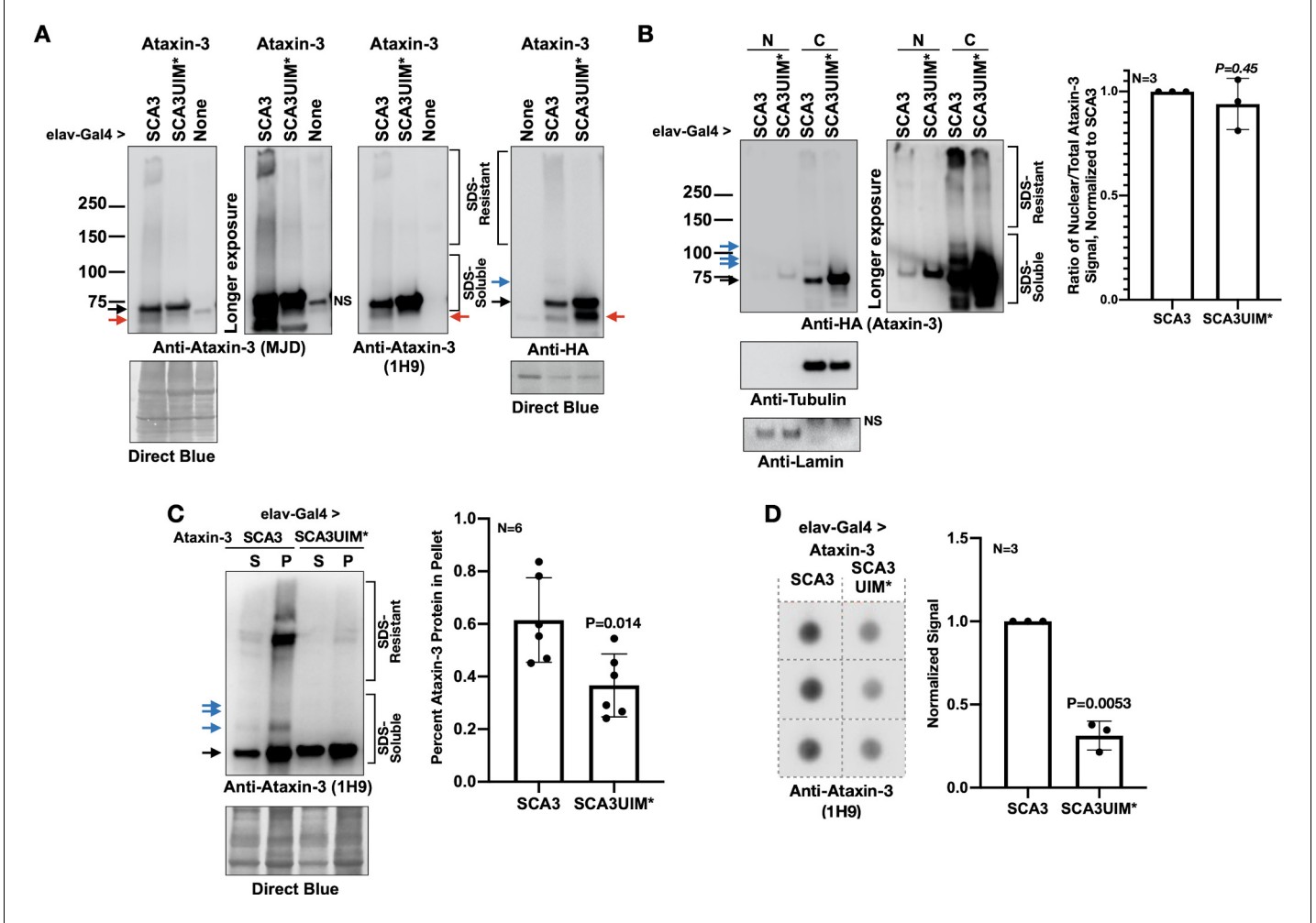

**Figure 4.** Mutating the UIMs of pathogenic ataxin-3 decreases its aggregation. (A) Western blots from simple lysates of adults expressing transgenes pan-neuronally. Red arrows: likely proteolytic products of ataxin-3 that we observe sometimes. (B) Cytoplasmic/nuclear fractionation of lysates from adults expressing transgenes pan-neuronally. Histograms: quantification of images from the left and other, independent biological replicates. Means -/+ SD. P value: two-tailed Student's t-test. Italicized P value denotes lack of statistical significance. The entire ataxin-3 signal in each lane was used for calculations, from the main band to the top. (C) Soluble/pellet centrifugation of lysates from adults expressing the noted transgenes. Histograms are from the left and other, independent biological repeats. Shown are means -/+ SD. P value: two-tailed, Student's t-test. The entire ataxin-3 signal in each lane was used for calculations, from the main band to the top. (D) Filter-trap assay of lysates from adult flies expressing transgenes pan-neuronally. Each image is from an independent biological repeat. Histograms: quantification of images from left. Means -/+ SD. P value: two-tailed, Student's t-test. Black arrows in panels: main ataxin-3 band. Blue arrows in panels: ubiquitinated ataxin-3. SDS-soluble and -resistant labels are based on prior work with ataxin-3 (*Sutton et al., 2017*; *Tsou et al., 2015a*; *Johnson et al., 2019*; *Todi et al., 2009*; *Todi et al., 2010*; *Tsou et al., 2013*).

The online version of this article includes the following source data for figure 4:

**Source data 1.** Data related to *Figure 4* - Numerical data.
**Source data 2.** Data related to *Figure 4* - Uncropped images.

## Hsc70-4 enhances aggregation and toxicity of pathogenic ataxin-3

Hsc70-4 belongs to the heat shock protein 70 (HSP70) superfamily. Its closest human ortholog is the constitutively expressed, heat shock protein family A member 8 (HSPA8; *Takayama et al., 1999*). Alongside other HSP70 members, its primary role is that of an ATP-dependent chaperone for unfolded proteins in protein quality control (*Yamamoto et al., 2010*; *Grove et al., 2011*; *Goodwin et al., 2014*; *Li et al., 2017*; *Sopha et al., 2012*; *Robert et al., 2019*; *Loeffler, 2019*; *Abildgaard et al., 2020*; *Davis et al., 2020*; *Faust and Rosenzweig, 2020*). HSPA8 is a part of the

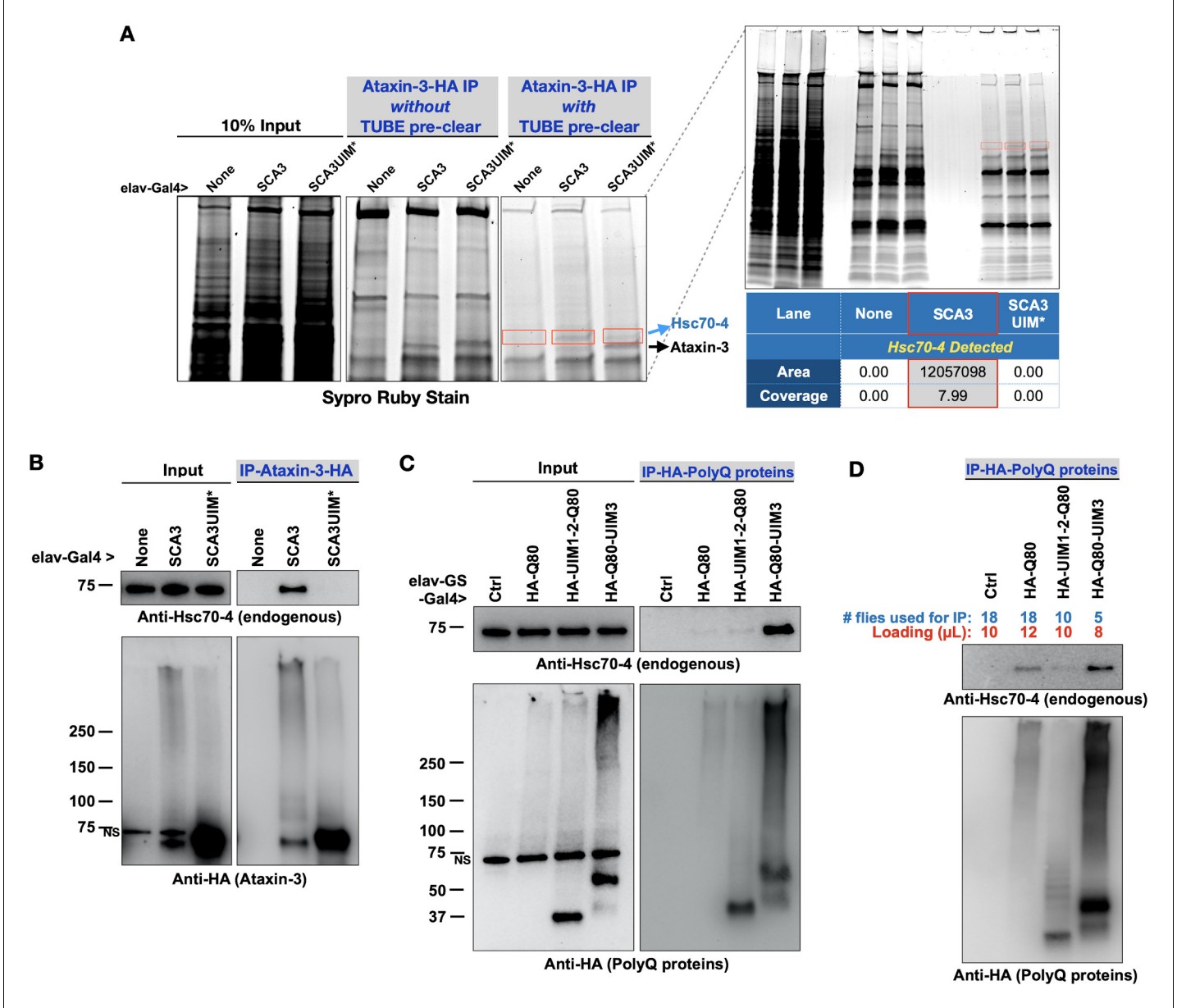

**Figure 5.** Pathogenic ataxin-3 interacts with Hsc70-4 in a UIM-dependent manner. (**A**) SDS-PAGE gel stained for all proteins with Sypro Ruby. Boxed areas were excised and proteins were identified with LC/MS/MS. Two versions of immunopurifications (IPs) were conducted. Middle panel shows results from IP-ed ataxin-3 from whole fly lysates. Right panel shows results from IP-ed ataxin-3 after lysates were first incubated with tandem ubiquitin binding entities (TUBEs; see the main text and Materials and methods). (**B–D**) Western blots from anti-HA IPs from flies. Results are representative of independent biological replicates conducted three times. NS: non-specific.

The online version of this article includes the following source data for figure 5:

**Source data 1.** Data related to *Figure 5* - Uncropped images.

ubiquitin-proteasome degradation system and is also involved in chaperone-mediated autophagy (*Robert et al., 2019*; *Loeffler, 2019*; *Stricher et al., 2013*; *Kampinga and Bergink, 2016*). Chaperones are presumed to alleviate toxicity from polyQ proteins by refolding or degrading them. Our studies, however, counterintuitively hint at the possibility that Hsc70-4 might do the opposite with ataxin-3, enhancing its polyQ toxicity.

Hsc70-4 is a different member of the HSP70 superfamily than fly Hsc70Cb and human HSPA1L, both of which were previously reported to suppress polyQ-dependent degeneration (*Warrick et al., 1999*; *Zhang et al., 2010*; *Iijima-Ando et al., 2005*; *Chan et al., 2002*). Also, there is recent

evidence that HSP70 members have pleiotropic effects on misfolded proteins (*Serlidaki et al., 2020*). Therefore, we knocked down Hsc70-4 in fly eyes to examine the possibility that it exacerbates SCA3. We selected fly eyes because widespread targeting of Hsc70-4 in the fly was developmentally lethal (larval, pupal, and pharate adult lethality), whereas expression in fly eyes was not problematic (clearly present pseudopupil and good overall morphology). We utilized three independent RNAi lines for Hsc70-4. Expression of each line in fly eyes led to reduced Hsc70-4 levels (*Figure 6A*). Levels of Hsc70-4 in fly eyes were likely even lower than what was captured by blots; we expressed RNAi in eyes but conducted blotting using whole heads that also contained tissues where Hsc70-4 was not targeted.

Next, we determined the effect of knocking down Hsc70-4 on the phenotype caused by pathogenic ataxin-3 by scoring fly eyes, as exemplified in *Figure 6B*; a higher number denotes a worse phenotype. Hsc70-4 knockdown consistently improved eye phenotype, denoted by the persistence of the pseudopupil − an indicator of underlying structure integrity − and fewer cases with depigmentation (*Figure 6C*). We also observed biochemical changes in ataxin-3 when Hsc70-4 was knocked down. Both SDS-resistant ataxin-3 species and aggregates isolated by filter-trap assays were reduced in the presence of Hsc70-4 RNAi (*Figure 6D,E*). Since aggregation of ataxin-3 is a key determinant of its toxicity in flies (*Ristic et al., 2018*; *Sutton et al., 2017*; *Tsou et al., 2015a*; *Johnson et al., 2019*), suppression of eye phenotypes by Hsc70-4 knockdown is likely a consequence of reduced SCA3 protein aggregation. Lastly, we tested the effect of Hsc70-4 knockdown on eye phenotypes from ataxin-3 with mutated UIMs. Knocking down Hsc70-4 did not have a detectable impact on eye phenotype caused by pathogenic ataxin-3(UIM*) (*Figure 6F*). Collectively, these data highlight Hsc70-4 as an enhancer of pathogenic ataxin-3 toxicity in a manner dependent on UIMs.

## Hsc70-4 knockdown suppresses polyQ toxicity more generally

We were intrigued by the suppressive effect of Hsc70-4 knockdown on pathogenic ataxin-3. Because we also observed Hsc70-4 co-IP with the isolated polyQ (*Figure 5C,D*) we tested whether its knockdown impacts polyQ80 eye phenotypes. Expression of polyQ80 led to pseudopupil loss and depigmentation. Knockdown of Hsc70-4 improved depigmentation, even though it did not recover the pseudopupil, indicative of ameliorated pathology (*Figure 7A*). Importantly, improved eye phenotype was accompanied by reduced polyQ80 aggregates (*Figure 7B*). (We note here that we have not observed a single, non-aggregated band of polyQ80 by western blots; we only observed aggregated species.)

We confirmed a protective effect from Hsc70-4 knockdown in yet another polyQ model (*Warrick et al., 1998*). This model contains polyQ78 and the C-terminus of an ataxin-3 isoform that lacks UIM3; UIMs 1 and 2 are also absent. Because expression of this polyQ species does not always yield a clear external eye phenotype, we examined internal structures. Normally, the ommatidia (eye units) are organized in a fan-like pattern, which was absent in the presence of polyQ78 (ATXN3$^{TR}$-Q78; *Figure 7C*). When Hsc70-4 was knocked down, the ommatidia were visible in a clear sign of toxicity suppression. Altogether, these results suggest the possibility of a wider role for Hsc70-4 as an enhancer of toxicity from different polyQ proteins.

## UIMs are important for inter-ataxin-3 interactions

So far, we observed that Hsc70-4 affects aggregation and toxicity of pathogenic ataxin-3 and that it interacts with the SCA3 protein through UIMs (*Figures 5* and *6*). Since our data also implicated Hsc70-4 in the toxicity of an isolated polyQ that does not contain UIMs (*Figure 7*) we wondered whether the UIMs of ataxin-3 have additional properties that might impact SCA3 protein aggregation.

Ataxin-3 proteins bind each other (*Ellisdon et al., 2006*; *Todi et al., 2007a*; *Masino et al., 2004*; *Masino et al., 2011*). Therefore, we investigated whether the UIMs play a part in inter-ataxin-3 associations. We examined whether HA-tagged, pathogenic ataxin-3 with intact or mutated UIMs co-IPs V5-tagged pathogenic ataxin-3 with intact UIMs in *Drosophila*. We found that the UIMs were important, but not necessary, for these interactions; a reduced amount of full-length, V5-tagged pathogenic ataxin-3 co-IP-ed with UIM* ataxin-3 compared to ataxin-3 with intact UIMs (*Figure 8A*). Additional co-IPs using truncated polyQ fragments and ataxin-3 with a wild-type polyQ indicated

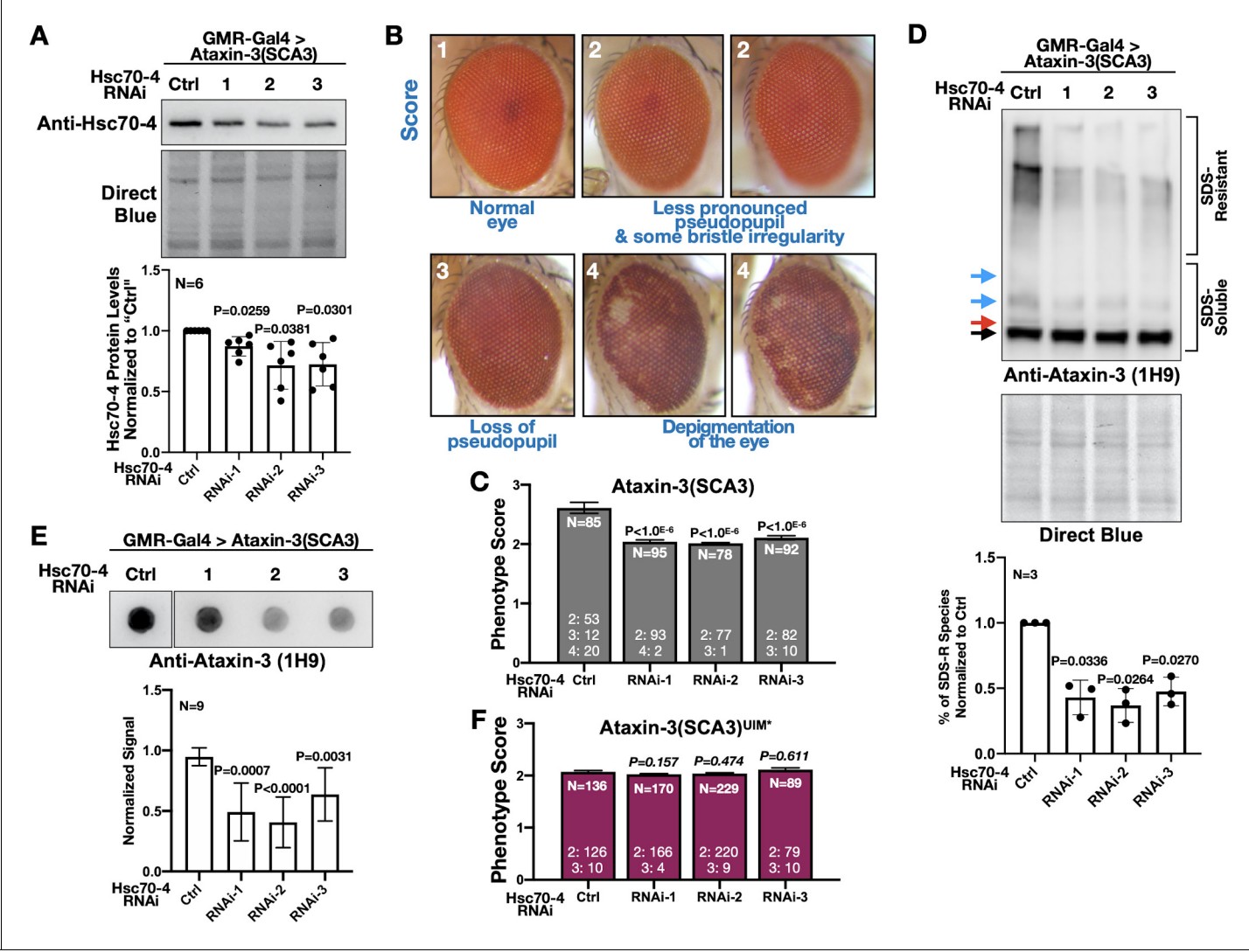

**Figure 6.** Hsc70-4 knockdown improves ataxin-3 toxicity in fly eyes. (A) Western blots from dissected fly heads. Histograms are from blots on top and other, independent biological repeats. Shown are means -/+ SD. Statistics: one-way ANOVA with Dunnett's correction. Lysates were from crosses used for studies in panels (B), (C) and other, similar experimental setups. (B) Representative images and summary of the scoring system designed to evaluate the effect of Hsc70-4 knockdown on the phenotype caused by pathogenic ataxin-3 in fly eyes. 1: normal-looking eyes that have a clearly defined pseudopupil and good overall appearance; 2: eyes where the pseudopupil is not as clearly pronounced compared with normal eyes and with some minor unevenness/irregularity in peripheral bristle arrangement; 3: eyes where the pseudopupil is not visible; 4: eyes with depigmentation. Quantitative outcomes are shown in panels (C) intact, pathogenic ataxin-3 and (F) UIM-mutated, pathogenic ataxin-3. (C) Statistics: Kruskal-Wallis tests comparing Hsc70-4 RNAi to Ctrl from independent biological replicates when pathogenic ataxin-3 is expressed in fly eyes. Shown are means -/+ SEM. (D) Western blots from dissected fly heads. Quantifications are from panels above and independent biological repeats. Shown are means -/+ SD. P values: one-way ANOVA with Dunnett's correction. Black arrow: unmodified ataxin-3. Red arrow: potentially phosphorylated species that we observe infrequently. Blue arrows: ubiquitinated ataxin-3. (E) Filter-trap assays. P values: one-way ANOVA with Dunnett's correction. Shown are means -/+ SD. Images are from the same membrane, cropped and reorganized for clarity. Full lane is in *Figure 6—figure supplement 1*. (F) Quantification of eye scores as in panels (B) and (C). Shown are means -/+ SEM. 'N's signify biological replicates. Italicized P values are not statistically significant. For panels (C) and (F), the numbers at the bottom of each histogram show the number of flies in each scoring category for that group.

The online version of this article includes the following source data and figure supplement(s) for figure 6:

**Source data 1.** Data related to *Figure 6* - Numerical data.
**Source data 2.** Data related to *Figure 6* - Uncropped images.
**Figure supplement 1.** Uncropped blot from panel 6E.

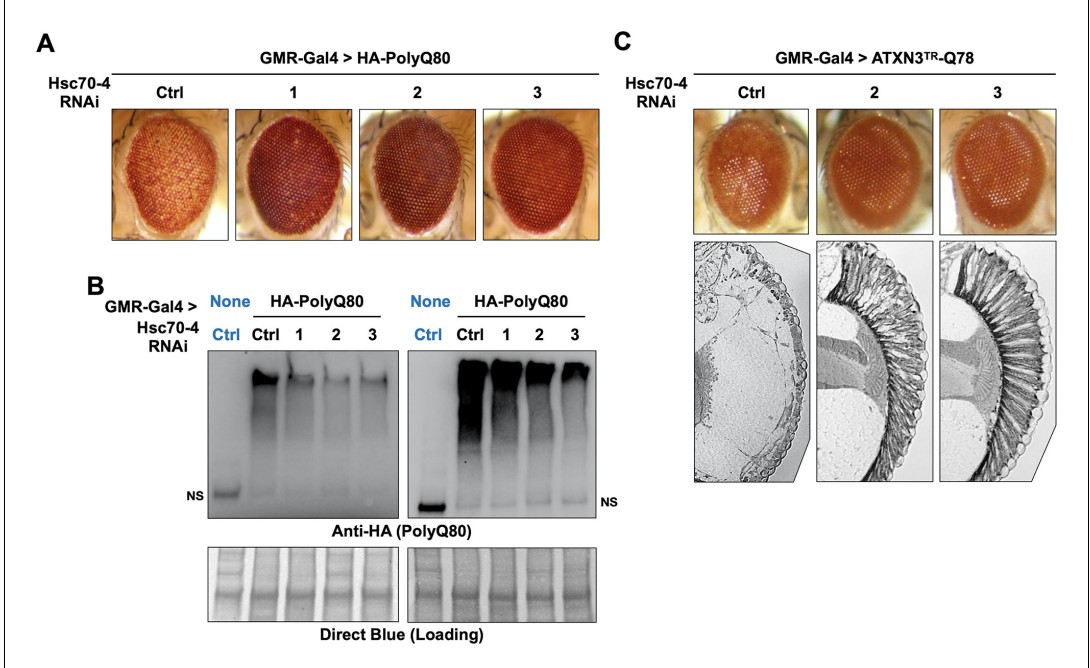

**Figure 7.** Hsc70-4 knockdown improves toxicity from independent, truncated polyQ models. Representative images of fly eyes (**A, C**) and fly eye sections (**C**) with the noted genotypes. Photos are representative of results from three independent sets of crosses. (**B**) Western blots from dissected heads. Results from two independent biological repeats are shown. 'Ctrl' in blue font did not express polyQ80. NS: non-specific.

The online version of this article includes the following source data for figure 7:

**Source data 1.** Data related to *Figure 7* - Uncropped images.

## Discussion

In this study, we systematically explored the role of protein context in SCA3 in an intact organism. We found that the UIMs of ataxin-3 are key players in pathogenicity through an unexpected role from Hsc70-4, which exacerbates SCA3. We propose that Hsc70-4 increases the toxicity of pathogenic ataxin-3 by interacting with its UIMs and worsening polyQ aggregation (*Figure 9A*). Additionally, since the UIMs facilitate inter-ataxin-3 binding, misfolded ataxin-3 proteins may be brought into close proximity that can further increase their chances of interaction, aggregation and toxicity (*Figure 9A*); as mentioned earlier, in our *Drosophila* studies of SCA3, ataxin-3 aggregation is consistently linked to its extent of pathogenicity. Our work provides the first, comprehensive view of the role of non-polyQ domains on ataxin-3 toxicity (*Figure 9*); it furthermore highlights an understudied and impactful property of a member of the normally beneficial HSP70 superfamily.

We previously reported on the role of three ataxin-3 domains on its polyQ toxicity: the catalytic domain, UbS2, and the VBM (*Figure 9B*). Based on our work in *Drosophila*, the catalytic domain is important for ataxin-3's ability to induce production of the co-chaperone, DnaJ-1, whose upregulation suppresses polyQ toxicity (*Sutton et al., 2017*; *Tsou et al., 2015a*). The capacity of ataxin-3 to upregulate DnaJ-1 also requires its binding to the proteasome-associated protein, Rad23 at UbS2 (*Sutton et al., 2017*; *Tsou et al., 2015a*). Through upregulated DnaJ-1, ataxin-3 suppresses its own polyQ-dependent toxicity (*Sutton et al., 2017*; *Tsou et al., 2015a*; *Tsou et al., 2013*). Therefore, it is not surprising that mutating either the catalytic site or UbS2 on pathogenic ataxin-3 renders the

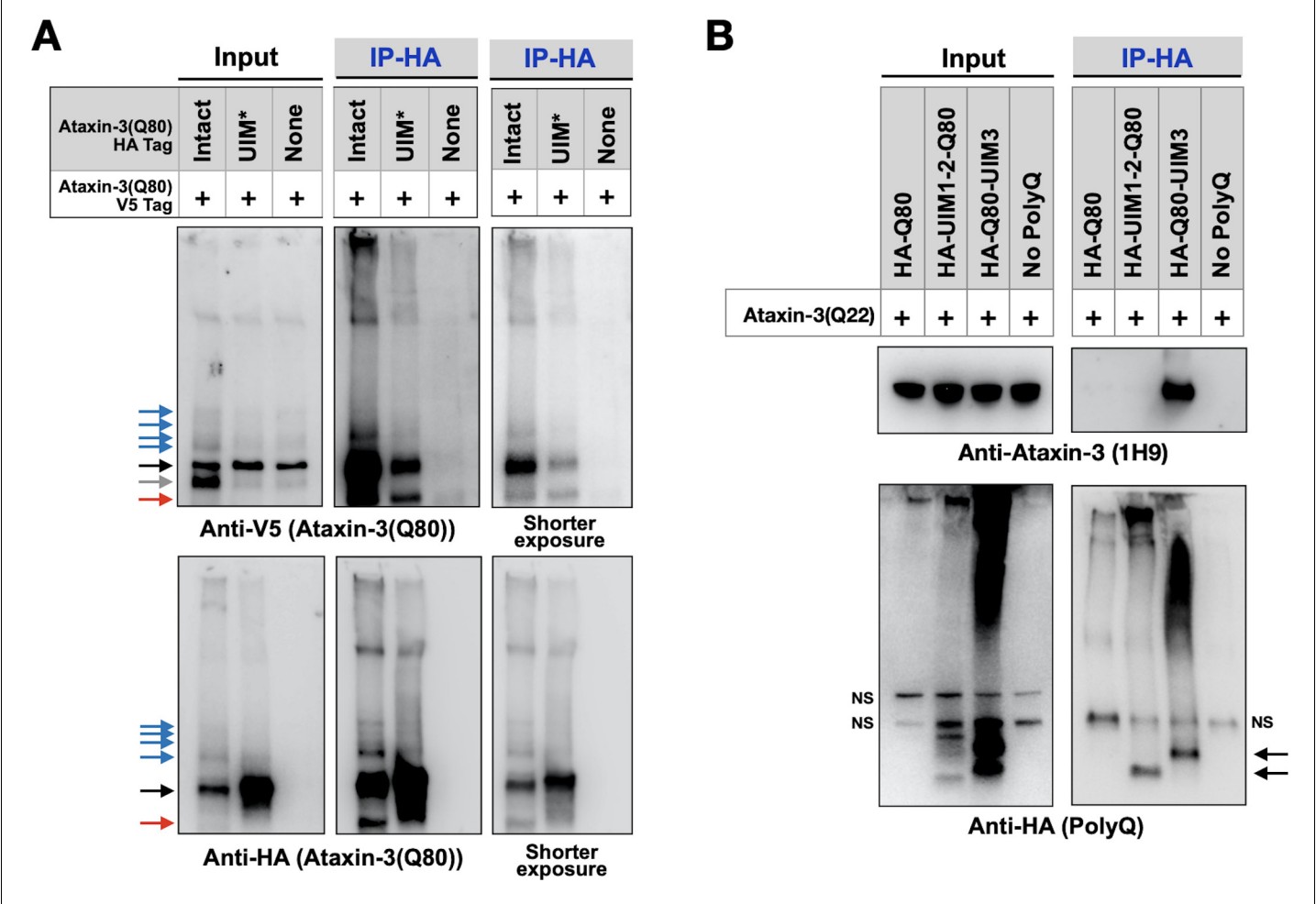

**Figure 8.** UIMs are important for inter-ataxin-3 interactions. (A, B) Western blots from IPs of ataxin-3 with expanded (A) or wild-type (B) polyQ repeats expressed in fly eyes. All transgenes (HA- or V5-tagged) were expressed at the same time and in the same tissues. Gray arrow: V5-positive signal absent in IP lanes. Red arrows: likely proteolytic fragments. Black arrows: unmodified proteins. Blue arrows: ubiquitinated ataxin-3. NS: non-specific. Images are representative of independent biological experiments conducted thrice.

The online version of this article includes the following source data for figure 8:

**Source data 1.** Data related to *Figure 8* - Uncropped images.

protein more toxic. The general suppressive contribution of each domain to ataxin-3 polyQ toxicity, as measured by longevity, is similar.

Unlike the catalytic site and UbS2, the other domains investigated — VBM and UIMs — act as enhancers of ataxin-3 polyQ toxicity. Elegant work by the Wanker and Bonini labs identified a region immediately preceding the polyQ of ataxin-3 as the binding site for VCP (*Boeddrich et al., 2006*; *Figure 9B*). The stoichiometry of this interaction was suggested to be four ataxin-3 proteins for one VCP hexamer (*Boeddrich et al., 2006*). These results and our findings that VCP binding enhances ataxin-3 aggregation and toxicity (*Ristic et al., 2018*) led us to propose that the interaction of multiple ataxin-3 proteins with a single VCP hexamer increases their chances of interaction and accelerates aggregation (*Ristic et al., 2018*; *Figure 9B*). The other domains that enhance the toxicity of ataxin-3 are the UIMs. Their relative impact on ataxin-3 polyQ toxicity might be stronger than the VBM (*Figure 9—figure supplement 1*). Our results strongly implicate Hsc70-4 as a driving force behind the role of the UIMs as enhancers of ataxin-3 pathogenicity.

It is intriguing that knockdown of Hsc70-4 reduces toxicity from pathogenic ataxin-3. As mentioned above, heat shock proteins generally work to improve toxicity from aggregated proteins (*Davis et al., 2020*; *Warrick et al., 1999*; *Duncan et al., 2015*; *Evans et al., 2010*;

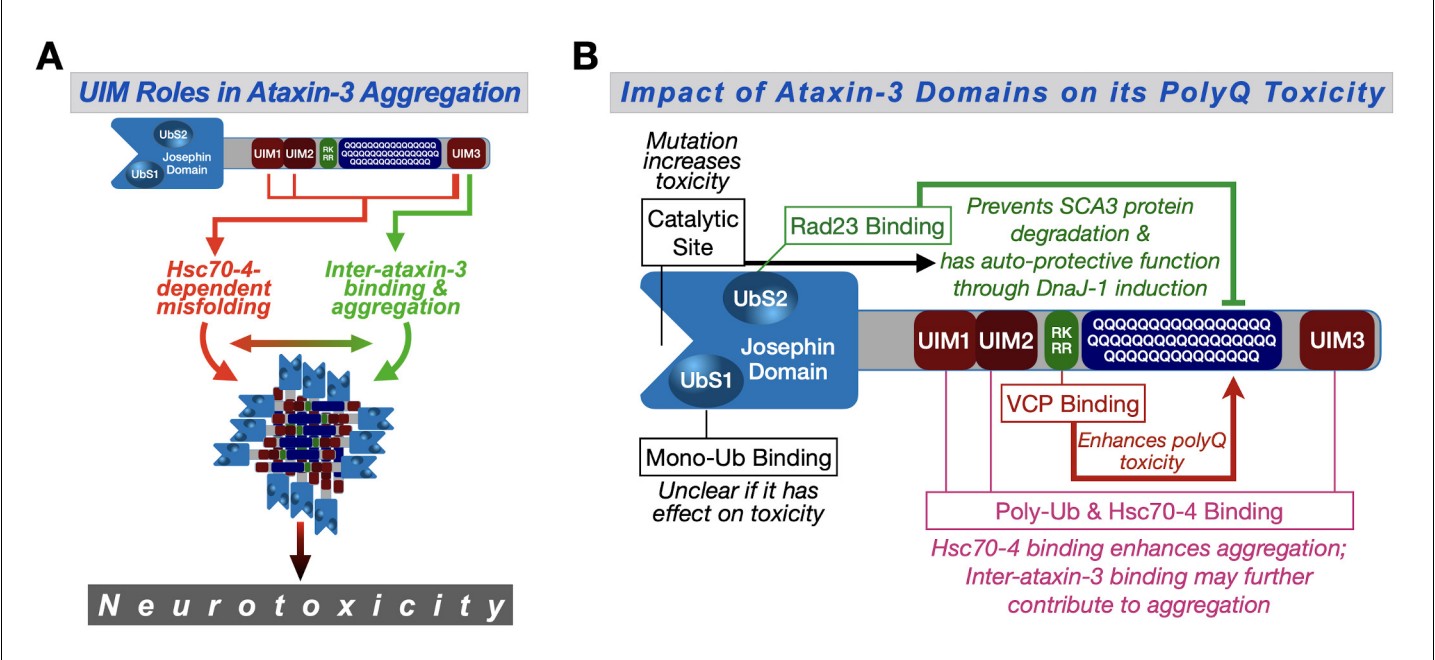

**Figure 9.** Proposed model. (**A**) Ataxin-3 UIMs enable its interaction with Hsc70-4 and other ataxin-3 proteins, both of which enhance aggregation and toxicity. UIM3 (thicker lines) appears to be a stronger contributor to these interactions than the other UIMs (thinner lines). (**B**) Summary of results from the current study and our prior publications on the effect of ataxin-3 domains on its polyQ toxicity (*Blount et al., 2014*; *Ristic et al., 2018*; *Sutton et al., 2017*; *Tsou et al., 2015a*; *Johnson et al., 2019*). Ub: ubiquitin.

The online version of this article includes the following figure supplement(s) for figure 9:

**Figure supplement 1.** Relative importance of UIMs on fly longevity.

*Thiruvalluvan et al., 2020*; *Chan et al., 2000*; *Vossfeldt et al., 2012*; *Kampinga and Bergink, 2016*). Hsc70-4 may act in a context-, protein-, and disease-dependent manner. For example, studies with its mammalian ortholog, HSPA8, indicate that it can suppress aggregation and toxicity of the HD-causing protein, huntingtin, harboring an expanded polyQ (*Scior et al., 2018*; *Monsellier et al., 2015*; *Bauer et al., 2010*). However, investigations with the disease-causing protein, Tau, suggest this chaperone as a potential enhancer of tauopathies. HSPA8 interacts with Tau and slows down its degradation; also, HSPA8 inhibition can reduce Tau levels in the brain (*Fontaine et al., 2015*; *Jinwal et al., 2009*; *Jinwal et al., 2013*). These data present the possibility of HSPA8 activities that can lead to enhanced toxicity from some proteins, similar to what we observed with ataxin-3.

How might Hsc70-4 enhance toxicity from pathogenic ataxin-3? Perhaps, ATP hydrolysis by Hsc70-4 is directly inhibited by binding to the UIMs of pathogenic ataxin-3, abrogating its refolding functions. It may also be that under normal circumstances ataxin-3 forms a functional complex with Hsc70-4, collaborating in protein quality control. Just like Hsc70-4/HSPA8, wild-type ataxin-3 is implicated in proteostasis (*Costa and Paulson, 2012*; *Dantuma and Herzog, 2020*). During normal conditions, a complex comprising wild-type ataxin-3, Hsc70-4 and other proteins could triage various substrates. When the polyQ of ataxin-3 is expanded, in a UIM-dependent manner, it may cause Hsc70-4 to adopt a dominant negative role and to enhance pathogenic ataxin-3 aggregation. Our study provides a roadmap toward testing these possibilities and uncovering others.

The model above focuses on the role of the UIMs facilitating an interaction between ataxin-3 and Hsc70-4, because pathogenic ataxin-3 with mutated UIMs does not co-IP Hsc70-4. However, we also observed that the isolated polyQ80 co-IP-ed some Hsc70-4 and that the chaperone's knockdown reduced its toxicity. Two earlier studies also reported the suppression of truncated polyQ toxicity by the knockdown of Hsc70-4 in flies (*Zhang et al., 2010*; *Vossfeldt et al., 2012*). (To the best of our knowledge, no additional findings have supplemented these data since their original publication.) Binding of Hsc70-4 to an isolated polyQ is not surprising. PolyQ peptides aggregate and heat shock

proteins interact with misfolded proteins (*Duncan et al., 2015*; *Evans et al., 2010*; *Kampinga and Bergink, 2016*). However, it is compelling that reducing levels of Hsc70-4 improves, instead of worsening, truncated polyQ toxicity. Other proteins involved with refolding, such as the HSP40 members Dnaj and mrj, and the HSP70 members Hsc70Cb and HSPA1L reduce polyQ protein aggregation and toxicity (*Sutton et al., 2017*; *Tsou et al., 2015a*; *Tsou et al., 2015b*; *Warrick et al., 1999*; *Zhang et al., 2010*; *Chan et al., 2000*; *Bilen and Bonini, 2007*). But these proteins are not the same as Hsc70-4/HSPA8, which may function differently depending on protein identity and cellular circumstances. In fact, HSPA8 is not the only member of the HSP70 superfamily to enhance the aggregation of a misfolded disease protein. Recent work in cultured mammalian cells indicated that HSPA1L — previously reported as a polyQ suppressor (*Warrick et al., 1999*; *Iijima-Ando et al., 2005*; *Chan et al., 2002*) — can in fact increase the aggregation of a non-polyQ misfolded protein, SOD1; the mechanism behind this effect is not entirely clear but it appears to involve HSPA1L binding partners (*Serlidaki et al., 2020*). Consequently, it is not outside of reasonable biological expectations that not all HSP70 members, such as Hsc70-4, function similarly; they may trigger different outcomes on misfolded proteins depending on their binding partners, the identity of the misfolded protein, the cellular environment and physiological conditions.

How might Hsc70-4 worsen polyQ fragment toxicity? The interaction of Hsc70-4 with severely truncated polyQ peptides could inhibit its ability to refold them, especially if additional proteins or binding partners are necessary to steer Hsc70-4 toward refolding, as suggested by studies of other HSP70 members (*Serlidaki et al., 2020*). This point pertains to isolated polyQ fragments. In the case of full-length ataxin-3, the UIMs' interaction with Hsc70-4 appears stronger and likely supersedes binding through the polyQ, leading to the models of aggregation postulated above. We draw this conclusion based on our results that mutating the UIMs of full-length, pathogenic ataxin-3 seems to abrogate its interaction with endogenous Hsc70-4. Additionally, knockdown of Hsc70-4 does not significantly change eye phenotype scores when the UIMs of full-length ataxin-3 are mutated. Nevertheless, we cannot fully discount the possibility that Hsc70-4 activities on the SCA3 protein may also depend on polyQ-based interactions, once UIM-based binding is established. At this point, the primary takeaway is that Hsc70-4 can enhance polyQ toxicity, in the case of ataxin-3 through its UIMs.

In summary, we demonstrated that non-polyQ domains of disease-causing ataxin-3 are key regulators of its toxicity, and that the UIMs are important determinants of the SCA3 protein's pathogenicity. Our studies establish a clear role for protein context in SCA3 and, through Hsc70-4, provide a unique entry point into further examinations and potential therapeutic solutions for this incurable ataxia. Lastly, our findings that Hsc70-4 can enhance polyQ toxicity have broader implications for the general understanding of chaperone biology.

# Materials and methods

## Antibodies

Anti-ataxin-3 (mouse monoclonal 1H9, MAB5360, 1:500–1000; Millipore), anti-MJD (rabbit polyclonal, 1:15,000 *Paulson et al., 1997*), anti-HA (rabbit monoclonal C29F4, 1:500–1000; Cell Signaling Technology), anti-V5 (mouse monoclonal R960-25, 1:500–1000; ThermoFisher), anti-tubulin (mouse monoclonal T5168, 1:10,000; Sigma-Aldrich), anti-lamin (mouse monoclonal ADL84.12–5, 1:1000; Developmental Studies Hybridoma Bank), anti-HSPA8/Hsc70-4 (rabbit monoclonal D12F2, 1:1000; Cell Signaling Technology), peroxidase conjugated secondary antibodies (goat anti-mouse, goat anti-rabbit, 1:5000–10,000; Jackson Immunoresearch).

## *Drosophila* materials and procedures

Flies were housed at 25°C in diurnal environments on conventional cornmeal or RU486-containing media. Common stocks were from Bloomington *Drosophila* Stock Center: GMR-Gal4 (#8121); isogenic host strain attP2 (#36303); Hsc70-4 RNAi lines #1, 2, and 3 (#28709, #34836, and #35684, respectively), GMR-QF2W (#59283), and ATXN3$^{TR}$-Q78 (#8141). Gifts included: sqh-Gal4 (Dr. Daniel Kiehart, Duke University), elav-GS-Gal4 (Dr. R. J. Wessells, Wayne State University), elav-Gal4 and repo-Gal4 (Dr. Daniel Eberl, University of Iowa). All flies were heterozygous for driver and transgene.

Ataxin-3 cDNAs were based on sequences from previous publications (*Ristic et al., 2018*; *Sutton et al., 2017*; *Tsou et al., 2015a*; *Johnson et al., 2019*; *Harris et al., 2010*; *Winborn et al.,*

*2008*; *Todi et al., 2009*; *Nicastro et al., 2010*; *Berke et al., 2005*; *Nicastro et al., 2009*). These include ataxin-3(SCA3): -intact, -catalytically inactive, -UbS2 mutated, -VBM mutated; as well as isolated polyQ80, UIM1-2-Q80, Q80-UIM3, UIM1-2-Q80-UIM3, and ataxin-3(SCA3)-UIMs mutated. Transgenes were sub-cloned into pWalium-10.moe. Transgenic lines were generated via phiC31 integration into attP2 on chromosome 3 (*Ristic et al., 2018*; *Sutton et al., 2017*; *Tsou et al., 2015a*; *Johnson et al., 2019*; *Tsou et al., 2016*). An additional ataxin-3 line was generated for *Figure 8* with the same sequence as in *Figure 1A*, but with a V5 tag, using plasmid pQUASp (Addgene #46162) and driven by GMR-QF2W. All transgene insertions were validated by PCR and genomic sequencing and western blotting, using procedures described before (*Ristic et al., 2018*; *Sutton et al., 2017*; *Tsou et al., 2015a*; *Johnson et al., 2019*; *Tsou et al., 2016*; *Blount et al., 2018*).

For longevity and motility, adults were collected on the day of eclosion. Deaths were recorded daily. Motility was tested weekly. For motility, adults were transferred into fresh vials 1 hr before assessment. Then, the percentage of flies per vial to reach the top at 5, 15, and 30 s was recorded after flies were forced to the bottom.

## Western blotting and quantification

Unless otherwise specified, 3 or 5 flies (depending on experiment), or 10 dissected adult heads per group were homogenized in boiling lysis buffer (50 mM Tris pH 6.8, 2% SDS, 10% glycerol, 100 mM dithiothreitol), sonicated, boiled for 10 min, and centrifuged at 13,300xg at room temperature for 10 min. Western blots were developed using PXi 4 (Syngene), or ChemiDoc (Bio-Rad). Blots were quantified with GeneSys (Syngene), or ImageLab (Bio-Rad), respectively. For direct blue staining, PVDF membranes were submerged for 10 min in 0.008% Direct Blue 71 (Sigma-Aldrich) in 40% ethanol and 10% acetic acid, rinsed in 40% ethanol/10% acetic acid, air dried, and imaged.

## Filter-trap assay

Three adult flies, or 10 dissected heads per group were homogenized in 200 µL NETN buffer (50 mM Tris, pH 7.5, 150 mM NaCl, 0.5% Nonidet P-40) supplemented with protease inhibitor cocktail (PI; S-8820, Sigma-Aldrich). Lysates were diluted with 200 µL PBS containing 0.5% SDS, sonicated briefly, then centrifuged at 4500×g for 1 min. 100 µL supernatant was diluted with 400 µL PBS. Thirty or 70 µL (depending on experimental setup) of each sample was filtered-vacuumed using Bio-Dot (Bio-Rad) through a 0.45 µm nitrocellulose membrane (Schleicher and Schuell) that was pre-incubated with 0.1% SDS in PBS. Membrane was rinsed twice with 0.1% SDS/PBS and analyzed by western blotting.

## Soluble/pellet centrifugation

Ten whole flies per group were lysed in 300 µL NETN buffer with PI, sonicated, then centrifuged at 20,000×g at 4℃ for 30 min. Supernatant was quantified with the BCA assay (ThermoFisher). Pellet was resuspended in 200 µL of PBS/1% SDS. Thirty µg of supernatant fraction and 7 µL of resuspended pellet was each supplemented with 6× SDS, boiled, and loaded for western blotting.

## Nuclear/cytoplasmic separation

Fractionation was performed using the ReadyPrep Protein Extraction Kit (Bio-Rad) using five whole flies per group that were lysed in cytoplasmic extraction buffer (Bio-Rad). Three times as much nuclear fraction was loaded onto gels compared to cytoplasmic fraction to eliminate the need for over-exposure.

## Co-immunopurifications

Fifteen whole flies or 5-30 dissected fly heads per group, depending on experiment, were lysed in NETN+PI, tumbled at 4℃ for 30 min, then centrifuged for 5 min at 10,000×g at 4℃. Supernatant was incubated with bead-bound antibody for 2–4 hr. Then, beads were rinsed 5× with NETN+PI. Bead-bound complexes were eluted by boiling in Laemmli buffer.

## Mass spectrometry

Forty whole flies per group were homogenized in 1 mL NETN+PI, sonicated, centrifuged at 4°C for 5 min at 10,000×g and supernatant was transferred into a fresh microfuge tube. Half of each sample was incubated with anti-HA beads (ThermoFisher) for 2 hr, rinsed 4× with NETN and bead-bound proteins were eluted with Laemmli buffer. The other half was tumbled for 30 min with an equal mixture of tandem ubiquitin binding entities (TUBEs) 1 and 2 (Lifesensors), centrifuged to isolate TUBE beads and the supernatant was incubated for 2 hr with anti-HA beads, rinsed and eluted as above. Eluates were resolved on an SDS-PAGE gel that was stained with Sypro Ruby (ThermoFisher). Bands of interest were excised and examined by LC/MS/MS at Wayne State University Mass Spectrometry Core.

## Histological preparation

The proboscises and wings of adult flies were removed before fixing overnight in 2% glutaraldehyde/2% paraformaldehyde in Tris-buffered saline/0.1% Triton X-100. Fixed bodies were dehydrated in a series of 30, 50, 70, and 100% ethanol for 1 hr each, washed in propylene oxide overnight, embedded in Poly/Bed812 (Polysciences), sectioned at 5 μm, and stained with toluidine blue.

## Statistical analyses

Statistical tests are specified in figure legends. Log-rank tests with Holm-Bonferroni adjustments, ANOVA, and Kruskal-Wallis tests were conducted in Prism 8 (GraphPad); Student's t-tests were conducted in Excel (Microsoft) or Numbers (Apple). P values are noted as reported by the software used. The number of biological replicates is noted on figures and corresponding legends. We have also noted in legends whether two-tailed or one-tailed tests were used. Two-tailed tests were used where a directionality of change was not necessarily predicted. One-tailed tests were used where a specific directionality in change between control and experimental groups was expected. In figures, non-statistically significant outcomes are noted in italicized font, whereas statistically significant outcomes are in normal font.

# Acknowledgements

We thank Dr. Matt Scaglione (Duke University) for critical input. We also thank Jamie Roebuck (Duke University Model System Genomics) for the generation of transgenic fly lines, and Madeline Somers (University of Michigan) for statistical consulting.

# Additional information

### Funding

| Funder | Grant reference number | Author |
|---|---|---|
| National Institute of Neurological Disorders and Stroke | R01NS08677 | Sokol V Todi |
| Wayne State University | Competitive Graduate Research Assistantship | Sean L Johnson |

The funders had no role in study design, data collection and interpretation, or the decision to submit the work for publication.

### Author contributions

Sean L Johnson, Conceptualization, Data curation, Software, Formal analysis, Funding acquisition, Validation, Investigation, Visualization, Methodology, Writing - original draft, Writing - review and editing; Bedri Ranxhi, Data curation, Validation, Investigation, Methodology, Writing - original draft; Kozeta Libohova, Data curation, Formal analysis, Supervision, Validation, Investigation, Visualization, Methodology; Wei-Ling Tsou, Conceptualization, Data curation, Software, Formal analysis, Supervision, Validation, Investigation, Visualization, Methodology, Writing - original draft, Writing - review and editing; Sokol V Todi, Conceptualization, Resources, Data curation, Software, Formal analysis,

Supervision, Funding acquisition, Validation, Investigation, Visualization, Methodology, Writing - original draft, Project administration, Writing - review and editing

## Author ORCIDs
Bedri Ranxhi  http://orcid.org/0000-0003-2508-731X
Sokol V Todi  https://orcid.org/0000-0003-4399-5549

## Decision letter and Author response
Decision letter https://doi.org/10.7554/eLife.60742.sa1
Author response https://doi.org/10.7554/eLife.60742.sa2

# Additional files

## Supplementary files
• Transparent reporting form

## Data availability
No large-scale datasets were generated or analyzed for this study. All results pertaining to the figures and the text are included in figures and supplementary files.

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
