## [Decision Letter]

**Acceptance summary:**

This is an interesting manuscript describing manipulation of Ataxin-3 UIMs in fly models of SCA3. Following up their own prior work, the authors present good evidence that UIMs, which comprise a substantial fraction of Ataxin-3, modulate Ataxin-3 polyQ toxicity. These findings are notable as they indicate how specific features of expanded polyQ containing proteins produce distinct phenotypes. In a counter-intuitive finding, a key actor appears to be interactions with the chaperone protein HSC-70-4. Overall, the work is well done, well presented, and well illustrated.

**Decision letter after peer review:**

Thank you for submitting your article "Ubiquitin-interacting motifs of ataxin-3 regulate its polyglutamine toxicity through Hsc70-4-dependent aggregation" for consideration by *eLife*. Your article has been reviewed by two peer reviewers, and the evaluation has been overseen by K VijayRaghavan as the Senior and Reviewing Editor. The following individuals involved in review of your submission have agreed to reveal their identity: Roger Albin (Reviewer #1); Udai Pandey (Reviewer #2).

The reviewers have discussed the reviews with one another and the Reviewing Editor has drafted this decision to help you prepare a revised submission.

Summary:

This is an interesting manuscript describing manipulation of Ataxin-3 UIMs in fly models of SCA3. Following up their own prior, nice work, the authors present good evidence that UIMs, which comprise a substantial fraction of Ataxin-3, modulate Ataxin-3 polyQ toxicity. These findings are notable as they indicate how specific features of expanded polyQ containing proteins produce distinct phenotypes. In a counter-intuitive finding, a key actor appears to be interactions with the chaperone protein HSC-70-4. Overall, the work is well done, well presented, and will illustrated.

Essential revisions:

I have one significant concern:

1) The authors also describe HSC-70-4 depletion as ameliorating pure polyQ toxicity. They have good data that UIM interactions are important in HSC-70-4 effects on the full transgene but how don't they know both mechanisms are operative.

2) The Q80-UIMs appears to be highly toxic than the QIM1-2-Q80-UIM3s in Figure 2D but the Q80-UIMs Seems to be living longer than QIM1-2-Q80-UIM3s Figure 2E. Can the author provide an explanation for this? One would assume that the more toxic a protein the more impact it should have on survival.

3) Since it is known that polyQ-expanded repeats sequester ubiquitin conjugates into aggregates, does mutating UIM (SCA3UIM*) decrease this since it is less aggregate prone in the two assays done in Figure 4? Does reducing Hsc70-4 have any effect on polyQ-expanded repeats-Mediated ubiquitin conjugates aggregation?

4) Figure 1—figure supplement 1 includes negative geotaxis assay that was never mentioned in the Results. The authors also mention in the Results "Also, there was no statistical difference between controls and lies expressing SCA3 (VBM*)" without mentioning what they are referring to in Figure 1—figure supplement 1, is it Figure 1—figure supplement 1A and B or both? Please make this clear.

5) What are the expression levels of the different construct in Figure 2D?

---

## [Author Response]

Essential revisions:I have one significant concern:1) The authors also describe HSC-70-4 depletion as ameliorating pure polyQ toxicity. They have good data that UIM interactions are important in HSC-70-4 effects on the full transgene but how don't they know both mechanisms are operative.

This is an important detail and we agree. We hope that the clarification below, in the Discussion, addresses this point.

"How might Hsc70-4 worsen polyQ fragment toxicity? […] Nevertheless, we cannot fully discount the possibility that Hsc70-4 activities on the SCA3 protein may also depend on polyQ-based interactions, once UIM-based binding is established. At this point, the primary takeaway is that Hsc70-4 can enhance polyQ toxicity, in the case of ataxin-3 through its UIMs."

2) The Q80-UIMs appears to be highly toxic than the QIM1-2-Q80-UIM3s in Figure 2D but the Q80-UIMs Seems to be living longer than QIM1-2-Q80-UIM3s Figure 2E. Can the author provide an explanation for this? One would assume that the more toxic a protein the more impact it should have on survival.

Thank you for this important detail. We believe that this comment is referring to Q80-UIM3. In the prior version of the manuscript we alluded to tissue-specific differences based on data from neuronal and glial cells. We have now further expanded on this point in the Results, with the following added text:

"There were tissue-dependent variations in toxicity from the above constructs. For example, Q80-UIM3 was markedly more toxic in fly eyes compared to the polyQ80 alone or with the addition of other UIMs (Figure 2D), whereas polyQ80 with all three UIMs was the most toxic species in adult neurons (Figure 2E). […] These outcomes highlight the utility of the new fly models that we have generated to understand tissue-selective toxicity in vivo in the future."

3) Since it is known that polyQ-expanded repeats sequester ubiquitin conjugates into aggregates, does mutating UIM (SCA3UIM*) decrease this since it is less aggregate prone in the two assays done in Figure 4? Does reducing Hsc70-4 have any effect on polyQ-expanded repeats-Mediated ubiquitin conjugates aggregation?

At present, we do not know the impact the UIMs might have on ataxin-3 inclusions and their decoration by endogenous ubiquitin. We know that ataxin-3 with mutated UIMs is itself less readily ubiquitinated (published work and other studies that we have not yet consolidated into a story; Todi et al., 2009; Todi et al., 2010). Whether this effect impacts the decoration of atxn3 aggregates is not clear at this time and is something we should investigate.

4) Figure 1—figure supplement 1 includes negative geotaxis assay that was never mentioned in the Results. The authors also mention in the Results "Also, there was no statistical difference between controls and lies expressing SCA3 (VBM*)" without mentioning what they are referring to in Figure 1—figure supplement 1, is it Figure 1—figure supplement 1A and B or both? Please make this clear.

We apologize for the lack of clarity in this part of the manuscript. We have amended it as follows in the Results:

"Next, we examined the effect of the same ataxin-3 transgenes in glial cells to obtain information on toxicity from pathogenic ataxin-3 in this cell type. […] There was no statistical difference between controls not expressing pathogenic ataxin-3 and flies expressing pathogenic ataxin-3 with mutated VBM both in terms of longevity (Figure 1—figure supplement 1A) and motility (Figure 1—figure supplement 1B). These findings highlight a need to explore the role of glia in SCA3."

5) What are the expression levels of the different construct in Figure 2D?

Overall levels are shown in Figure 2—figure supplement 1. Results are from whole, dissected fly heads. We caution here that the comparison of overall polyQ protein levels is not precise, especially concerning polyQ80-UIM3, which leads to markedly severe eye degeneration. The driving reason for generating these truncated lines was to conduct an initial exploration of the role of UIMs on the isolated polyQ of ataxin-3. The main focus of the present work is full-length ataxin-3 and the role of the UIMs in its pathogenicity, explored in subsequent figures. Future work will dissect how individual UIMs impact various properties of the isolated polyQ80 in a tissue- and age-dependent manner.